# Federated Learning Under Second-Order Data Heterogeneity

## Abstract

We consider the problem of Federated Learning over clients with heterogeneous data. We propose an algorithm called SABER that samples a subset of clients and tasks each client with its own local subproblem. SABER provably reduces client drift by incorporating an estimate of the global update direction and regularization into each client's subproblem. Under second-order data heterogeneity with parameter $\delta$, we prove that the method's communication complexity for non-convex problems is $\mathcal{O}\left(\delta \varepsilon^{-2} \sqrt{M}\right)$. In addition, for problems satisfying $\mu$-Polyak-Łojasiewicz condition, the method converges linearly with communication complexity of $\mathcal{O}\left(\left(\frac{\delta}{\mu}\sqrt{M} + M\right)\log\frac{1}{\varepsilon}\right)$. To showcase the empirical performance of our method, we compare it to standard baselines including FedAvg, FedProx, and SCAFFOLD on image classification problems and demonstrate its superior performance in data-heterogeneous settings.

## 1 Introduction

Federated learning (FL) has emerged as a distributed learning paradigm that enables user devices to collaboratively train a global model in a privacy-preserving manner. Since its advent (McMahan et al., 2017), a significant body of work has aimed at tackling key challenges that obstruct its broad deployability, with emphasis on the communication cost (Konečnỳ et al., 2016; Li et al., 2020c), the system diversity of client devices (Yang et al., 2021; Abdelmoniem et al., 2023a) and inconsistent accuracy across individual users (Yu et al., 2020; Wu et al., 2020). In this effort, techniques such as compressed communication (Reisizadeh et al., 2020; Hönig et al., 2022), adaptable model architectures (Horvath et al., 2021; Karimireddy et al., 2020), dynamic client selection (Lai et al., 2021; Li et al., 2022) and partial model personalization (Collins et al., 2021; Pillutla et al., 2022) have made significant strides towards actual FL deployments (Yang et al., 2018; Bonawitz et al., 2019; Paulik et al., 2021; Huba et al., 2022).

Despite the progress in various fronts, a central and still withstanding problem in FL is the heterogeneity of data across clients (Kairouz et al., 2021). Our main interest in this work is to design algorithms for settings with heterogeneous data, formalized by an assumption on *second-order data heterogeneity*. Before we proceed to a formal definition, let us discuss the intuition behind it. In many practical scenarios, the clients would have similar kinds of inputs, while their outputs may vary significantly (Silva et al., 2022). In general, this is a problem of clients having *preferences*, which makes the data very heterogeneous, and represents the setting that we are interested in. To quote Arivazhagan et al. (2019):

> "same input data can receive different labels from different users".

Kairouz et al. (2021) call this situation a *concept shift*. This problem is typically tackled using personalization (Arivazhagan et al., 2019). However, personalization only solved the data-heterogeneity issue when the devices have enough computation and memory budget to fine-tune the model. When targeting the out-of-the-box performance, non-iid data remain a big challenge. As pointed out by Jiang et al. (2019), global performance and personalized performance are often conflicting objectives, so in this paper, we consider the

non-personalized formulation given below:

$$\min_{\mathbf{w} \in \mathbb{R}^d} f(\mathbf{w}) = \frac{1}{M} \sum_{m=1}^{M} f_m(\mathbf{w}), \tag{1}$$

where the functions $f_1, \ldots, f_M$ are continuously differentiable. As a blanket assumption, we assume that the objective above is lower bounded by some finite value $f_* = \inf_{\mathbf{w}} f(\mathbf{w}) > -\infty$. Our main interest in this paper is in the case where the functions might be non-convex.

The most common approach to developing federated learning algorithms is to assume bounded data heterogeneity based on *first-order data heterogeneity*:

$$\|\nabla f_m(\mathbf{w}) - \nabla f(\mathbf{w})\| \leq \delta. \qquad \text{(first-order heterogeneity assumption)}$$

This assumption imposes a restriction on how dissimilar the gradients of different clients can be. It has been commonly used to study convergence of FedAvg and other gradient methods for federated learning (Wang et al., 2018; Yu et al., 2019a). It also follows from an even more restrictive yet popular (Li et al., 2020b; Yu et al., 2019b; Nguyen et al., 2022b) assumption of uniformly bounded gradients. While it has been shown to be unnecessary for studying the convergence of FedAvg (Khaled et al., 2020), FedAvg does not, in general, converge precisely to the optimum due to the client drift induced by data heterogeneity.

In our view, the first-order heterogeneity assumption is not well suited for non-iid settings, where clients optimize for different objectives. For instance, in logistic regression, the gradients can be pointing in the opposite directions if they have different labels. In contrast, *second-order heterogeneity* does not need the gradients to be correlated and, as we will discuss later, it is closer connected to the mere similarity of the inputs. But first, we give a formal definition below.

**Assumption 1.** *We assume that the data have bounded second-order heterogeneity with constant $\delta \geq 0$, meaning that for any $m \in \{1, \ldots, M\}$, it holds*

$$\|\nabla^2 f(\mathbf{w}) - \nabla^2 f_m(\mathbf{w})\| \leq \delta \quad \text{for all } \mathbf{w}.$$

Assumption 1 is also sometimes referred to as *Hessian similarity*. It can also be formulated without assuming twice differentiability, if for any $\mathbf{w}_1, \mathbf{w}_2 \in \mathbb{R}^d$ it holds

$$\|\nabla f(\mathbf{w}_1) - \nabla f_m(\mathbf{w}_1) - (\nabla f(\mathbf{w}_2) - \nabla f_m(\mathbf{w}_2))\| \leq \delta \|\mathbf{w}_1 - \mathbf{w}_2\|. \tag{2}$$

When all functions are twice-differentiable, the two formulations are equivalent. A proof of how Hessian similarity implies (2) can be found in the work of Khaled & Jin (2022). To the best of our knowledge, the assumption was first introduced by Mairal (2013) in the context of using surrogate losses with $m = 2$, and with a general $m \geq 2$ it was first considered by Shamir et al. (2014).

This assumption is satisfied for a wide range of problems where the input data are similar, whereas the labels do not matter as much. For instance, consider the regression task $f_m(\mathbf{w}) = \mathbb{E}_{(x_m, y_m) \sim \mathcal{D}_m}[\frac{1}{2}(n(\mathbf{w}; x_m) - y_m)^2]$, where $n(\mathbf{w}; \cdot)$ is a predictor, such as a neural network with weights $\mathbf{w}$, and $x_m, y_m$ are a random pair of input and label. Then, $\nabla^2 f_m(\mathbf{w}) = \mathbb{E}_{(x_m, y_m)}[\nabla^2 n(\mathbf{w}; x_m)(n(\mathbf{w}; x_m) - y_m) + \nabla n(\mathbf{w}; x_m) \nabla n(\mathbf{w}; x_m)^\top]$. Notice that the second part of the Hessian of $f_m$ is completely independent of $y_m$ and depends only on $x_m$. Furthermore, if $\mathbb{E}_{(x_m, y_m)}[\nabla^2 n(\mathbf{w}; x_m)(n(\mathbf{w}; x_m) - y_m)] = 0$, which holds, for example, for linear models, then $\nabla^2 f_m(\mathbf{w})$ only depends on the distribution of $x_m$, and $\delta \leq \max_{\mathbf{w}, m, m'} \|\mathbb{E}_{x_m, x_{m'}}[\nabla n(\mathbf{w}; x_m) \nabla n(\mathbf{w}; x_m)^\top - \nabla n(\mathbf{w}; x_{m'}) \nabla n(\mathbf{w}; x_{m'})^\top]\|$. Furthermore, as shown by Woodworth et al. (2023), this is not restricted to squared loss and the input similarity is sufficient for logistic regression as well. In other words, as long as different clients have similar distributions of inputs, they can have widely different outputs even for the same input, and $\delta$ will still be a small number. This is a significant improvement compared to the first-order heterogeneity assumption.

Our work might also be of interest outside of federated learning. Chayti & Karimireddy (2022) and Woodworth et al. (2023) provided a number of machine learning applications, where even having two functions can be useful for machine learning tasks. For example, when using simulators or synthetic data sources, the

Table 1: A summary of related work and conceptual differences to our approach.

| Algorithm | 2nd-order data heterogeneity | Non-Convex theory | Partial participation | Stateless | Reference |
|-----------|------------------------------|-------------------|----------------------|-----------|-----------|
| FedAvg | ✗ | ✓ | ✓ | ✓ | McMahan et al. (2017) |
| FedProx | ✗ | ✓ | ✓ | ✓ | Li et al. (2020a) |
| FedDANE | ✗ | ✓ | ✓ | ✓ | Li et al. (2019) |
| SCAFFOLD | ✓ | ✗[(1)] | ✗[(2)] | ✗ | Karimireddy et al. (2020) |
| SVRP | ✓ | ✗ | ✓ | ✓ | Khaled & Jin (2022) |
| SVRS | ✓ | ✗ | ✓ | ✓ | Lin et al. (2023) |
| SABER | ✓ | ✓ | ✓ | ✓ | **This work** |

[(1,2)] SCAFFOLD's theory under second-order data heterogeneity (Theorem IV in (Karimireddy et al., 2020)) is only for full participation and quadratic problems. The latter condition also implies that the global objective must be convex for a solution to exist.

data become heterogeneous and our method can be used to minimize an objective $f$ while having access to a number of proxies $f_1, \ldots, f_M$. For simplicity of presentation, we leave those applications out of consideration in our work and focus on federated learning.

We summarize our contributions as follows:

- We develop a new algorithm called SABER that is *stateless* by design and supports *partial participation*;

- We prove that it achieves state-of-the-art communication complexity of $\mathcal{O}\left(\delta \varepsilon^{-2} \sqrt{M}\right)$ on general non-convex problems;

- We prove a faster linear convergence with complexity $\mathcal{O}\left(\left(\frac{\delta}{\mu}\sqrt{M} + M\right) \log \frac{1}{\varepsilon}\right)$ when the objective is $\mu$-Polyak-Łojasiewicz, which improves upon the $\mathcal{O}\left(\left(\frac{\delta^2}{\mu^2} + M\right) \log \frac{1}{\varepsilon}\right)$ complexity of SVRP (Khaled & Jin, 2022) whenever $\frac{\delta}{\mu} \geq \sqrt{M}$, and matches the complexity of SVRS (Lin et al., 2023). Moreover, unlike SVRP and SVRS, our method does not require the objective to be convex;

- We run experiments on logistic regression and neural network problems and compare the performance of SABER to standard baselines such as FedProx and SCAFFOLD.

## 2   Related Work

**FL Methods for Data Heterogeneity.** Having observed that the attainable accuracy of the status-quo FedAvg algorithm (McMahan et al., 2017) degrades significantly under realistic data-heterogeneous setups (Luo et al., 2021), several works have attempted to alleviate it. Prominent existing methods have proposed either to define a regularization term that aims to discourage the local model from excessively deviating from the server model (Li et al., 2020a), or to introduce control variates, either for the whole model (Karimireddy et al., 2020; Mishchenko et al., 2022) or for part of it (Li et al., 2023), as a mechanism for estimating and compensating the drift between client and server models. The covariates can provably mitigate data heterogeneity (Mishchenko et al., 2022), but at the cost of requiring the clients to maintain a state. Departing from the standard FL goal of learning a single model, an alternative line of work has relaxed this setting by focusing on jointly learning multiple models, either per client (Smith et al., 2017) or per group of clients (Briggs et al., 2020; Sattler et al., 2021).

Shamir et al. (2014) were among the first to work on second-order data heterogeneity and proposed a stateless method called DANE, which is limited to full client participation and quadratic problems. Li et al. (2019) proposed a generalization of DANE called FedDANE that supports partial participation and general non-convex objectives, but their analysis does not consider second-order data heterogeneity. The analyses of both FedDANE (Li et al., 2019) and FedProx (Li et al., 2020a) rely on a more restrictive first-order data heterogeneity assumption.

A more recent line of work considered methods that are both stateless and work under second-order data heterogeneity, such as the aforementioned SVRP (Khaled & Jin, 2022) and SVRS (Lin et al., 2023). SVRP is motivated by approximating the SVRG update (Johnson & Zhang, 2013) with proximal updates, which can be solved locally by the clients. SVRS, as explained by Lin et al. (2023), can be seen as a modification of SVRP that uses gradient sliding (Lan, 2016) at each iteration. Unlike SVRP and our method, SVRS picks one of the functions, $f_1(\mathbf{w})$, to denote the loss on extra data stored on the server, which should approximate the overall objective $f(\mathbf{w})$, and use $f_1$ to construct a subproblem at each iteration. In contrast to SVRP, which requires each $f_m$ to be strongly convex, SVRS only requires strong convexity of $f$, and it has better complexity on ill-conditioned problems. We, in turn, do not assume strong convexity at all and study non-convex problems. Khaled & Jin (2022) also proposed to accelerate SVRP using the Catalyst framework of Lin et al. (2015), and Lin et al. (2023) did the same thing for SVRS too. Since Catalyst is agnostic to the method used as a subsolver, it can be applied to SABER as well. However, it would most likely result in the same convergence rate as that of Catalyzed SVRP or SVRS, and on top of that, it is unlikely to give a practical method, so we do not explore this direction.

**Orthogonal FL Techniques.** In an endeavor to improve other aspects of FL, existing efforts have also focused on alleviating the communication cost and the system heterogeneity across clients. On the communication efficiency front, existing work has attempted to minimize the exchanged data either through compression methods, such as top-$K$ sparsification (Rothchild et al., 2020) and various quantization schemes (Reisizadeh et al., 2020; Hönig et al., 2022), or by communicating variable amount of data as the training process progresses, with techniques such as progressive pruning (Jiang et al., 2022) or growing (Alam et al., 2022) of the model. With respect to client-side system heterogeneity, existing methods include resource-aware client selection schemes (Lai et al., 2021; Li et al., 2022; Abdelmoniem et al., 2023b), elastic model architectures whose complexity can be scaled based on the processing capabilities of each client device, with respect to arithmetic precision (Yoon et al., 2022) or in the width (Diao et al., 2020; Horvath et al., 2021), depth (Liu et al., 2022) or both width and depth dimensions (Ilhan et al., 2023), and more exotic methods, such as ZeroFL's sparse convolutions (Qiu et al., 2021) and FedBuff's buffered asynchronous aggregation (Nguyen et al., 2022a). The majority of these methods are orthogonal to our work and can be combined with SABER with varying amount of effort.

## 3 A new method

Here, we propose SABER, a new *stateless* method that tries to estimate the batch gradient $\nabla f(\mathbf{w}_k)$ using an accumulated sequence $\mathbf{v}_k$. Key components of SABER comprise *i)* a new local objective function, which allows us to maintain a single control variate $\mathbf{v}_k$ that is shared across clients, and *ii)* a control variate updating rule with a tunable synchronization interval. Concretely, we devise the term $\langle \mathbf{v}_k - \nabla f_m(\mathbf{w}_k), \mathbf{w} - \mathbf{w}_k \rangle$, which is added to the local objective function of each client. Unlike SCAFFOLD, which requires a per-client control variate, our approach performs bias correction by solely utilizing the current gradient on the given client $\nabla f_m(\mathbf{w}_k)$, which is readily available on each participating client through the use of gradient-based optimizers, and the shared vector $\mathbf{v}_k$. The complete local objective on the $m$-th client is derived as

$$f_m(\mathbf{w}) + \langle \mathbf{v}_k - \nabla f_m(\mathbf{w}_k), \mathbf{w} - \mathbf{w}_k \rangle + \frac{1}{2\eta} \|\mathbf{w} - \mathbf{w}_k\|^2,$$

where the first term is the loss of the client's target learning task, the second term performs bias correction, and the third term introduces regularization with the goal to limit client drift.

For our control variate update rule, we are motivated by the recursive estimators for stochastic optimization (Nguyen et al., 2017; Fang et al., 2018; Richtárik et al., 2021) that rely on gradient differences to improve

the current estimate, and MARINA (Gorbunov et al., 2021) is a particularly relevant method. Specifically, we selectively perform either an assignment to the average gradient of a newly sampled subset of clients $\widetilde{\mathcal{S}}_k$ with probability $p$ or a refinement of the current estimate using the already sampled subset of clients. Algorithm 2 presents the full details of SABER, including the local objective optimization (lines 13-14) and the control variate update rule (lines 5-11).

To position SABER with respect to prior work and understand why it works, let us look at the following four standard FL algorithms:

**FedAvg** works by simply running SGD locally on each client, and, thus, the local update can be written down simply as

$$\mathbf{w}_{k+1,m} \approx \arg \min_{\mathbf{w}} f_m(\mathbf{w}).$$

**FedProx** modifies the update to reduce the client drift using regularization parameterized by $\eta$:

$$\mathbf{w}_{k+1,m} \approx \arg \min_{\mathbf{w}} \left\{ f_m(\mathbf{w}) + \frac{1}{2\eta} \|\mathbf{w} - \mathbf{w}_k\|^2 \right\}.$$

**SCAFFOLD & Scaffnew** are both based on the idea of reducing the client drift more explicitly. In a nutshell, they incorporate a global direction $\mathbf{v}_k$ and the local drift $\mathbf{v}_{k,m}$ of client $m$ to then realign the update on client $m$ as follows:

$$\mathbf{w}_{k+1,m}^{i+1} = \mathbf{w}_{k+1,m}^i - \gamma(\nabla f_m(\mathbf{w}_{k+1,m}^i) + \mathbf{v}_k - \mathbf{v}_{k,m}),$$

where $i$ is the local iteration counter on client $m$. We can deduce the objective the client in SCAFFOLD is trying to minimize from this update rule by assuming that it converges to some value $\mathbf{w}_{k+1,m}$. If that is the case, then $\nabla f_m(\mathbf{w}_{k+1,m}) + \mathbf{v}_k - \mathbf{v}_{k,m} = 0$, which is the first-order optimality condition for the following subproblem:

$$\mathbf{w}_{k+1,m} \approx \arg \min_{\mathbf{w}} \left\{ f_m(\mathbf{w}) + \langle \mathbf{v}_k - \mathbf{v}_{k,m}, \mathbf{w} - \mathbf{w}_k \rangle \right\}.$$

**SABER** can be seen as a combination of FedProx with SCAFFOLD/Scaffnew, although we emphasize that SABER was derived purely based on the assumption of data heterogeneity. The core of its update can be written as

$$\mathbf{w}_{k+1,m} \approx \arg \min_{\mathbf{w}} \left\{ f_m(\mathbf{w}) + \langle \mathbf{v}_k - \mathbf{v}_{k,m}, \mathbf{w} - \mathbf{w}_k \rangle + \frac{1}{2\eta} \|\mathbf{w} - \mathbf{w}_k\|^2 \right\}.$$

As we can see, the new subproblem includes a bias-correction term that is similar to the one in SCAFFOLD, and a regularization term just as in FedProx. The way we define $\mathbf{v}_k$ and $\mathbf{v}_{k,m}$ is, however, quite different from that of SCAFFOLD.

Our update rules are motivated by the recursive estimators for stochastic optimization (Nguyen et al., 2017; Fang et al., 2018; Richtárik et al., 2021) that use gradient differences to improve the current estimate. The main idea behind recursive estimates is to maintain a vector $\mathbf{v}_k$ that estimates the full gradient, i.e., $\mathbf{v}_k \approx \nabla f(\mathbf{w}_k)$. The previous approximation is updated with the following recursive formula:

$$\mathbf{v}_k = \mathbf{v}_{k-1} + \nabla f_{m_k}(\mathbf{w}_k) - \nabla f_{m_k}(\mathbf{w}_{k-1}).$$

It is easy to show this improves the expectation of the estimate vector since

$$\mathbb{E}[\mathbf{v}_k] = \mathbb{E}[\mathbf{v}_{k-1} + \nabla f(\mathbf{w}_k) - \nabla f(\mathbf{w}_{k-1})] = \mathbb{E}[\nabla f(\mathbf{w}_k)].$$

The main difference with SARAH, however, is how we use this estimate. In SARAH, it is used to directly update the parameters:

$$\mathbf{w}_{k+1} = \mathbf{w}_k - \eta \mathbf{v}_k, . \tag{SARAH}$$

Since $\mathbf{v}_k$ approximates $\nabla f(\mathbf{w}_k)$, SARAH's update, thus, approximates the update of gradient descent. We, however, do not assume $f$ to be a smooth function, and, therefore, we want to approximate the implicit proximal-point update (Moreau, 1965) instead, which requires to have $\mathbf{w}_{k+1} \approx \mathbf{w}_k - \eta \nabla f(\mathbf{w}_{k+1})$. This is

---

**Algorithm 1** SABER (**S**tochastic **A**ccumulated **B**atch-gradient **E**stimato**R**) - simplified version

---

1: **Input:** initialization $\mathbf{w}_0 \in \mathbb{R}^d$, stepsize $\eta > 0$, probability of synchronization $p > 0$
2: $\mathbf{w}_{-1} = \mathbf{w}_0$, $\mathbf{v}_0 = \nabla f(\mathbf{w}_0)$
3: **for** $k = 0, 1, 2, \ldots$ **do**
4:     Sample $m_k$ uniformly from $\{1, \ldots, M\}$
5:     $\mathbf{v}_k = \begin{cases} \nabla f(\mathbf{w}_k), & \text{with probability } p \\ \mathbf{v}_{k-1} + \nabla f_{m_k}(\mathbf{w}_k) - \nabla f_{m_k}(\mathbf{w}_{k-1}) & \text{otherwise} \end{cases}$
6:     Construct local objective $\phi_k(\mathbf{w}) = f_{m_k}(\mathbf{w}) + \langle \mathbf{v}_k - \nabla f_{m_k}(\mathbf{w}_k), \mathbf{w} - \mathbf{w}_k \rangle + \frac{1}{2\eta} \|\mathbf{w} - \mathbf{w}_k\|^2$
7:     Find $\mathbf{w}_{k+1} \approx \operatorname{argmin}_{\mathbf{w}} \phi_k(\mathbf{w})$
8: **end for**

---

where we use the insights from the work of Woodworth et al. (2023) and construct a subproblem to produce the new point:

$$\mathbf{w}_{k+1} \approx \arg\min_{\mathbf{w}} \left\{ f_{m_k}(\mathbf{w}) + \langle \mathbf{v}_k - \nabla f_{m_k}(\mathbf{w}_k), \mathbf{w} - \mathbf{w}_k \rangle + \frac{1}{2\eta} \|\mathbf{w} - \mathbf{w}_k\|^2 \right\}.$$

By first-order optimality conditions for the subproblem, we have

$$\mathbf{w}_{k+1} \approx \mathbf{w}_k - \eta(\mathbf{v}_k + \nabla f_{m_k}(\mathbf{w}_{k+1}) - \nabla f_{m_k}(\mathbf{w}_k)).$$

Even though the full expectation of $\nabla f_{m_k}(\mathbf{w}_{k+1})$ is not equal to $\nabla f(\mathbf{w}_{k+1})$ due to dependence between $\mathbf{w}_{k+1}$ and $m_k$, we can still notice that intuitively this update gives $\mathbb{E}[\mathbf{w}_{k+1}] \approx \mathbf{w}_k - \eta \nabla f(\mathbf{w}_{k+1})$. Woodworth et al. (2023) have a slightly different approach and they do not use any estimates such as our $\mathbf{v}_k$, which is why their non-convex theory guarantees first-order stationarity only up to a variance term.

PAGE (Li et al., 2021) and FedPAGE (Zhao et al., 2021) use the same principle to produce variance-reduced methods, the latter of which was designed for federated learning. The key difference is that they operate under $L$-smoothness of the objective and their complexity depends on $L$ rather than $\delta$. Since $\delta \leq 2L$, our approach is slightly more general, and it will result in a faster convergence whenever $\delta$ is significantly smaller than $L$.

Our theory covers convergence of Algorithm 1 in the general non-convex setting as well as under an extra assumption called Polyak-Łojasiewicz condition. In practice, it is common to use batches of clients instead of a single client as given in Algorithm 2. To keep the theory simple, we only study Algorithm 1 since the effect of minibatching has already been thoroughly studied (Gower et al., 2019; Khaled & Richtárik, 2020).

### 3.1 Solving the subproblem

Our theory requires that we minimize the bias-corrected regularized local objective. In practice, this can be achieved with any solver such as SGD or Adam that would otherwise work on the same problem without the extra terms. In fact, the regularization term only makes the problem easier to solve since it convexifies the objective.

From the theoretical point of view, we can characterize the difficulty of solving the subproblem when introducing extra assumptions. For instance, if $f_m$ is $L$-smooth, i.e., its gradient is $L$-Lipschitz, then whenever $\eta \leq \frac{1}{L}$, the subproblem becomes convex. Furthermore, as shown by Woodworth et al. (2023), it is usually enough to find an approximate stationary point, i.e., get $\mathbb{E}[\|\nabla \phi_k(\mathbf{w})\|^2] \leq \varepsilon$. If regularization does not make the objective convex, the (stochastic) gradient complexity of Algorithm 1 is not better than for solving the original problem. The communication complexity, on the other hand, improves substantially as we show in the next section.

**Assumption 2** (Inexact solution)**.** *We assume that $\mathbf{w}_{k+1} \approx \operatorname{argmin}_{\mathbf{w}} \phi_k(\mathbf{w})$ in the sense that the following two conditions hold:*

    *1. Monotonicity: $\mathbb{E}[\phi_k(\mathbf{w}_{k+1})] \leq \phi_k(\mathbf{w}_k)$.*

---

**Algorithm 2** SABER - full version

---

1: **Input:** initialization $\mathbf{w}_0 \in \mathbb{R}^d$, stepsize $\eta > 0$, probability of synchronization $p > 0$
2: $\mathbf{w}_{-1} = \mathbf{w}_0$, $\mathbf{v}_{-1} = \mathbf{v}_0 = \nabla f(\mathbf{w}_0)$
3: **for** $k = 0, 1, 2, \ldots$ **do**
4:      Sample a subset of clients $\mathcal{S}_k$
5:      Sample a Bernoulli variable $b_t \in \{0, 1\}$ with $\mathbb{P}(b_t = 1) = p$
6:      **if** $b_t == 1$ **then**
7:          Sample a new subset of clients $\widetilde{\mathcal{S}}_k$
8:          $\mathbf{v}_k = \frac{1}{|\widetilde{\mathcal{S}}_k|} \sum_{j \in \widetilde{\mathcal{S}}_k} \nabla f_j(\mathbf{w}_k)$
9:      **else**
10:          $\mathbf{v}_k = \mathbf{v}_{k-1} + \frac{1}{|\mathcal{S}_k|} \sum_{m \in \mathcal{S}_k} (\nabla f_m(\mathbf{w}_k) - \nabla f_m(\mathbf{w}_{k-1}))$
11:      **end if**
12:      **for** client $m \in \mathcal{S}_k$ **do**
13:          $\phi_{k,m}(\mathbf{w}) = f_m(\mathbf{w}) + \langle \mathbf{v}_k - \nabla f_m(\mathbf{w}_k), \mathbf{w} - \mathbf{w}_k \rangle + \frac{1}{2\eta} \|\mathbf{w} - \mathbf{w}_k\|^2$
14:          $\mathbf{w}_{k+1,m} \approx \operatorname{argmin}_{\mathbf{w}} \phi_{k,m}(\mathbf{w})$
15:      **end for**
16:      $\mathbf{w}_{k+1} = \frac{1}{|\mathcal{S}_k|} \sum_{m \in \mathcal{S}_k} \mathbf{w}_{k+1,m}$
17: **end for**

---

  2. *Almost stationarity:* $\mathbb{E}[\|\nabla \phi_k(\mathbf{w}_{k+1})\|^2] \leq \varepsilon$.

## 3.2 Convergence theory

Our theory for Algorithm 1 is comprised of several building blocks. First of all, we need a descent-like property, which is given in the next lemma.

**Lemma 1.** Define $\mathbf{w}_{k+1}$ as in Algorithm 1, and let $\eta \leq \frac{1}{4\delta}$, then

$$f(\mathbf{w}_{k+1}) - f(\mathbf{w}_k) \leq -\frac{1}{4\eta} \|\mathbf{w}_{k+1} - \mathbf{w}_k\|^2 + 2\eta \|\mathbf{v}_k - \nabla f(\mathbf{w}_k)\|^2. \tag{3}$$

As suggested by Lemma 1, the loss can be minimized as long as $\mathbf{v}_k$ stays sufficiently close to $\nabla f(\mathbf{w}_k)$. Therefore, we need to control the norm of the approximation error, which we do in the next lemma.

**Lemma 2.** The iterates of Algorithm 1 satisfy

$$\mathbb{E}\left[\|\mathbf{v}_{k+1} - \nabla f(\mathbf{w}_{k+1})\|^2\right] \leq (1 - p)\|\mathbf{v}_k - \nabla f(\mathbf{w}_k)\|^2 + \delta^2 \mathbb{E}\left[\|\mathbf{w}_{k+1} - \mathbf{w}_k\|^2\right].$$

Lemma 1 and Lemma 2 together suggest that the loss should decrease proportionally to $\frac{1}{\eta}\|\mathbf{w}_{k+1} - \mathbf{w}_k\|^2$. This property on its own, however, is not sufficient to show that we converge to a reasonable fixed point, it might merely mean that the method stops making progress. The last lemma that we need shows that the method actualy keeps making progress as long as the gradient norm is large.

**Lemma 3.** If $\eta \leq \frac{1}{4\delta}$ and $\mathbf{w}_{k+1}$ satisfies Assumption 2, then it holds

$$\mathbb{E}\left[\|\mathbf{w}_{k+1} - \mathbf{w}_k\|^2\right] \geq \eta^2 \mathbb{E}\left[\frac{1}{5}\|\nabla f(\mathbf{w}_{k+1})\|^2 - \|\mathbf{v}_k - \nabla f(\mathbf{w}_k)\|^2 - \varepsilon\right]. \tag{4}$$

**Theorem 1.** *Consider the iterates of Algorithm 1 and assume Assumptions 1 and 2 hold. If $\eta \leq \frac{\sqrt{p}}{4\delta}$, then it holds*

$$\frac{1}{K} \sum_{k=1}^{K} \mathbb{E}\left[\|\nabla f(\mathbf{w}_k)\|^2\right] \leq \frac{20(f(\mathbf{w}_0) - f_*)}{\eta K} + 5\varepsilon.$$

*Proof sketch.* The proof idea is based on defining the following Lyapunov function with a constant $c > 0$:

$$\mathcal{L}_k \stackrel{\text{def}}{=} f(\mathbf{w}_k) + c\|\mathbf{v}_k - \nabla f(\mathbf{w}_k)\|^2.$$

We show in the proof of Theorem 1 in the appendix that $\mathcal{L}_k$ satisfies a simple recursion:

$$\mathbb{E}\left[\mathcal{L}_{k+1}\right] \leq \mathcal{L}_k - \frac{cp}{6}\|\mathbf{v}_k - \nabla f(\mathbf{w}_k)\|^2 - \frac{\eta}{16}\mathbb{E}\left[\|\nabla f(\mathbf{w}_{k+1})\|^2\right]. \tag{5}$$

To derive this property, we simply need to combine Lemma 1 with Lemma 2 and Lemma 3. Since we have error terms in all lemmas, we need to balance the terms, which we achieve by assuming $\eta \leq \frac{\sqrt{p}}{4\delta}$. $\qquad \square$

One can notice that with higher value of $p$, we get a larger term subtracted from the right-hand side. Moreover, increasing $p$ also allows us to increase the stepsize $\eta$, which makes sense since we get a more accurate accumulated-gradient estimate when $p$ is larger.

If we set $p = \frac{1}{M}$ and $\eta = \frac{\sqrt{p}}{4\delta}$, we obtain a point $\mathbf{w}_K$ satisfying $\mathbb{E}\left[\|\nabla f(\mathbf{w}_K)\|^2\right] \leq \varepsilon$ after $K = \mathcal{O}\left(\frac{\varepsilon^{-2}}{\eta}\right) = \mathcal{O}\left(\sqrt{M}\delta\varepsilon^{-2}\right)$ communication rounds.

### 3.3 Improved convergence rate under PŁ assumption

To compare the complexity of SABER to that of other works, we also need to establish a linear convergence under extra assumptions. However, unlike previous papers, we do not want to use strong convexity since it would defeat the main point of extending the theory to non-convex functions. Instead, we use the PŁ assumption given below.

**Assumption 3.** *We say that $f$ satisfies Polyak-Łojasiewicz property if for any $\mathbf{w} \in \mathbb{R}^d$ it holds*

$$\frac{1}{2}\|\nabla f(\mathbf{w})\|^2 \geq \mu(f(\mathbf{w}) - f_*). \tag{6}$$

Let us slightly modify the Lyapunov function to increase the coefficient of the functional values:

$$\Psi_k \stackrel{\text{def}}{=} \mathcal{L}_k - f_* + \frac{\eta\mu}{8}(f(\mathbf{w}_k) - f_*) = (1 + \eta\mu/8)f(\mathbf{w}_k) - f_* + c\|\mathbf{v}_k - \nabla f(\mathbf{w}_k)\|^2.$$

Assuming that $f$ is $\mu$-PŁ, we then get

$$\mathbb{E}\left[\Psi_{k+1} - \frac{\eta\mu}{8}(f(\mathbf{w}_{k+1}) - f_*)\right] \stackrel{(5)}{\leq} \Psi_k - \frac{\eta}{16}\mathbb{E}\left[\|\nabla f(\mathbf{w}_{k+1})\|^2\right] - \frac{cp}{6}\|\mathbf{v}_k - \nabla f(\mathbf{w}_k)\|^2$$

$$\leq \Psi_k - \frac{\eta\mu}{8}\mathbb{E}\left[f(\mathbf{w}_{k+1}) - f_*\right] - \frac{cp}{6}\|\mathbf{v}_k - \nabla f(\mathbf{w}_k)\|^2.$$

Rearranging the terms, we obtain

$$\mathbb{E}\left[\Psi_{k+1}\right] \leq \max\left(1 - \frac{\eta\mu}{8}, 1 - \frac{p}{6}\right)\Psi_k.$$

.

This implies the following theorem.

**Theorem 2.** *Assume that the objective is $\mu$-PŁ. Then, to achieve $\mathbb{E}\left[\Psi_K\right] = \mathcal{O}(\varepsilon)$, Algorithm 1 requires at most $K = \mathcal{O}\left(\left(\frac{1}{\eta\mu} + \frac{1}{p}\right)\log\frac{1}{\varepsilon}\right)$ communication rounds. In particular, if we set $p = \frac{1}{M}$ and $\eta = \frac{1}{4\sqrt{p}\delta}$, then the complexity is $\mathcal{O}\left(\left(\frac{\delta}{\mu}\sqrt{M} + M\right)\log\frac{1}{\varepsilon}\right)$.*

As we mentioned previously, this complexity matches the one of SVRS (Lin et al., 2023) but ours does not need convexity.

### 3.4 Partial participation

With probability $p$, Algorithm 1 uses all clients to update $\mathbf{v}_k$. However, in many cases using all clients might be impossible, so in Algorithm 2 we consider updating $\mathbf{v}_k$ using only a subset of clients $\widetilde{\mathcal{S}}_k$. This setting can be studied under an extra assumption that the variance over clients is bounded:

**Assumption 4.** *For the setting of Algorithm 2 where only a subset of clients is used to update $\mathbf{v}_k$, i.e., $\mathbf{v}_k = \frac{1}{|\widetilde{\mathcal{S}}_k|} \sum_{m \in \widetilde{\mathcal{S}}_k} \nabla f_m(\mathbf{w}_k)$, we also assume that the variance in bounded:*

$$\mathbb{E}\left[\|\nabla f_m(\mathbf{w}_k) - \nabla f(\mathbf{w}_k)\|^2\right] \leq \sigma^2. \tag{7}$$

Under this assumption, we establish the following result.

**Theorem 3.** *Consider the same setting as in Theorem 1 with the update for $\mathbf{v}_k$ produced by subsampling clients, i.e., $\mathbf{v}_k = \frac{1}{|\widetilde{\mathcal{S}}_k|} \sum_{m \in \widetilde{\mathcal{S}}_k} \nabla f_m(\mathbf{w}_k)$. If $\eta \leq \frac{\sqrt{p}}{4\delta}$ and $|\mathcal{S}_k| = C$ for all $k$, then*

$$\frac{1}{K} \sum_{k=1}^{K} \mathbb{E}\left[\|\nabla f(\mathbf{w}_k)\|^2\right] \leq \frac{20(f(\mathbf{w}_0) - f_*)}{\eta K} + 5\varepsilon + 60\frac{\sigma^2}{C}.$$

The assumption that $|\mathcal{S}_k| = C$ for all $k$ is used in Theorem 3 only to simplify the statement, and a similar result would also hold if $|\mathcal{S}_k|$ varies over iterations. The main takeaway message from this additional result is that whenever there are sufficiently many clients participating in the update of $\mathbf{v}_k$, the last quantity is going to be small and so we do not need all clients to participate.

## 4 Experiments

We empirically evaluate our method on a logistic regression problem and on two different classification datasets, i.e., CIFAR-10 (Krizhevsky, 2009) and Federated EMNIST (FEMNIST) (Caldas et al., 2018). We compare the performance of SABER with baselines including FedAvg (McMahan et al., 2017), FedProx (Li et al., 2020a), and SCAFFOLD (Karimireddy et al., 2020).

### 4.1 Experimental Setup

For the empirical evaluation, we implemented our algorithm using PyTorch and integrated it into the Flower framework (Beutel et al., 2020) v1.2.0. We run all our experiments on NVIDIA RTX A6000 GPUs with CUDA version 11.7, Python 3.8.13, and PyTorch 1.13.1. The code will be open-sourced upon acceptance of this paper.

CIFAR-10 contains 60,000 32×32 color images in 10 different classes. The original dataset is split among 100 clients based on label distribution that are dissimilar from client to client to emulate data heterogeneity in realistic FL settings, as widely adopted in prior works (Horvath et al., 2021; Li et al., 2023). We split CIFAR-10 following Latent Dirichlet Allocation (LDA) partitioning with $\alpha = \{0.1, 1.0\}$, where lower value implies higher data heterogeneity. FEMNIST consists of 28×28 grey-scale images of digital hand-written alphabets and digits, hence 62 classes and naturally partitioned based on each client's handwriting.

For all baselines and the proposed SABER algorithm, we use ResNet-18 (He et al., 2016) for both CIFAR-10 and FEMNIST. For all methods, on each communication round, we randomly select 10 clients for model updates, using an SGD optimizer and a stepsize of 0.01, with batch size equal to 32. The clients perform 1 local training epoch on each round. For SABER specifically, we sample a subset of clients (100 for FEMNIST and 50 for CIFAR-10) for the control variate update with $p$=0.5. We use proximal coefficient $\eta$=0.5 for both SABER and FedProx.

### 4.2 Logistic Regression

We run experiments on binary classification with logistic regression loss. We target the 'w8a' and 'a9a' datasets from LIBSVM, which were also partitioned in a heterogeneous manner to 20 clients. We tune the

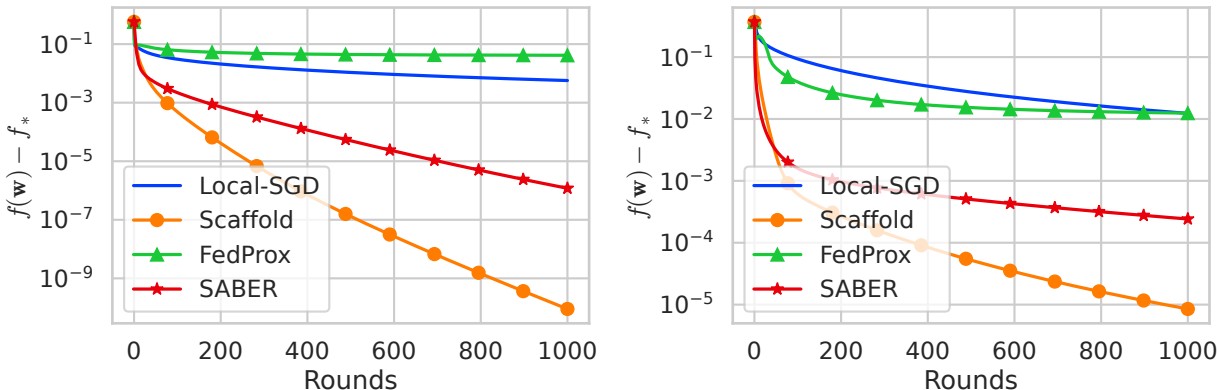

Figure 1: Convergence of SABER and other methods on logistic regression with full gradients. Left: 'w8a' dataset; right: 'a9a' dataset.

**Table 2:** Required number of communication rounds to achieve a target top-1 accuracy.

| Method | CIFAR-10 (62%) $\alpha = 0.1$ | CIFAR-10 (62%) $\alpha = 1.0$ | FEMNIST (70%) Non-iid |
|---|---|---|---|
| FedAvg | 841 (1×) | 331 (1×) | 66 (1×) |
| FedProx | 796 (1.06×) | 331 (1×) | 71 (0.93×) |
| SCAFFOLD | 1806 (0.47×) | 776 (0.43×) | 151 (0.44×) |
| **SABER (Ours)** | **446 (1.89×)** | **261 (1.27×)** | **56 (1.18×)** |

stepsize for each method individually and report the best configuration. All methods use full gradients and full participation of 20 clients in total. The results in Figure 1 show that SABER improves substantially upon Local-SGD, which is equivalent to FedAvg in this setting, and FedProx. SABER does not outperform SCAFFOLD in these specific experiments, despite having a much stronger theory. This can be attributed to the fact that we tuned the hyperparameters, so SCAFFOLD works better than predicted by its theory. Furthermore, SCAFFOLD is algorithmically similar to Scaffnew (Mishchenko et al., 2022), which has accelerated rate independently of data heterogeneity under the right choice of hyperparameters. Together, these observations explain why SCAFFOLD performed the best in these experiments.

## 4.3 Empirical Performance on Benchmark Datasets

To empirically investigate SABER's performance, we evaluate two setups: *i)* rounds-to-accuracy, which captures the communication rounds needed to reach a target accuracy, and *ii)* round-constrained accuracy, which shows the achieved accuracy under a budget of communication rounds.

Table 2 lists the achieved rounds-to-accuracy of each method for the target datasets and heterogeneity settings. The target top-1 accuracy is 62% for CIFAR-10 and 70% for FEMNIST. We observe that SABER achieves the lowest rounds-to-accuracy over all methods across all settings. Importantly, SABER yields speedups of 1.89×, 1.78×, and 4.04× over FedAvg, FedProx and SCAFFOLD, respectively, in the most data-heterogeneous case of CIFAR-10 with $\alpha = 0.1$.

To showcase the performance of SABER, we further look at the achieved top-1 accuracy at 1,000 and 400 communication rounds for CIFAR-10 and FEMNIST respectively, as shown in Table 3. Compared with the baselines, SABER performs at the same level on CIFAR-10 with $\alpha = 1.0$ and FEMNIST. Notably, as data heterogeneity increases on CIFAR-10 with $\alpha = 0.1$, our method reaches accuracy gains of 14.14 percentage points (pp), 8.3 pp and 18.2 pp over FedAvg, FedProx and SCAFFOLD, respectively, indicating its effectiveness in the presence of data heterogeneity.

**Table 3:** Top-1 accuracy (%) after 1,000 and 400 rounds for CIFAR-10 and FEMNIST, respectively.

| Method | CIFAR-10 $\alpha = 0.1$ | CIFAR-10 $\alpha = 1.0$ | FEMNIST Non-iid |
|---|---|---|---|
| FedAvg | 50.88 | 73.36 | 81.67 |
| FedProx | 56.72 | 73.56 | 80.88 |
| SCAFFOLD | 46.82 | 63.54 | 78.64 |
| **SABER (Ours)** | **65.02** | **73.88** | **82.50** |

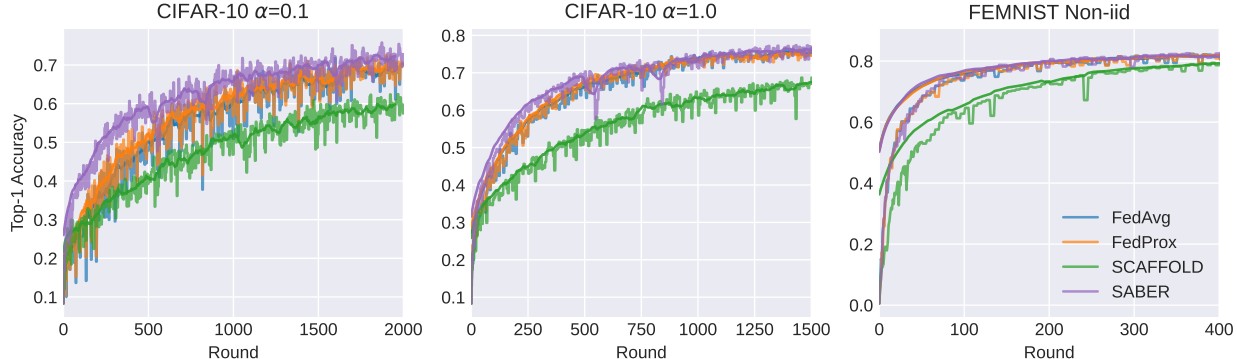

Figure 2: Validation accuracy recorded during training (real-time and 50 step running average) for FedAvg, FedProx, SCAFFOLD, and the proposed SABER on CIFAR-10 with LDA $\alpha = \{0.1, 1.0\}$ and the non-iid FEMNIST dataset.

Finally, Figure 2 shows the convergence behavior of the compared methods. Similarly to the previous evaluation setups, SABER provides the fastest convergence across all datasets, with higher gains under higher data heterogeneity.

## 5 Conclusion

In this paper, we propose SABER, an algorithm for federated learning under second-order data heterogeneity. SABER solves a local subproblem at each client, while mitigating client drift with a novel combination of *i)* estimation of global update direction and *ii)* regularization to the local objective. Theoretical study shows SABER achieves state-of-the-art communication complexity in non-convex problems. In practice, SABER is stateless by design, and it supports partial participation, making it a strong candidate for real-world deployment of federated learning. We showcase the performance of SABER on both logistic regression tasks, and non-convex deep learning setting of image classification on CIFAR-10 and FEMNIST datasets. When compared with standard baselines such as FedAvg, SABER achieves up to $1.89\times$ speedup and 14.4 pp gain in accuracy.

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

## A  Theory

### A.1  Proof of Lemma 1

*Proof.* Let $m = m_k$. By differentiability of $f$ and $f_m$, we have

$$f(\mathbf{w}_{k+1}) - f(\mathbf{w}_k) = \int_0^1 \langle \nabla f(\mathbf{w}_k + \tau(\mathbf{w}_{k+1} - \mathbf{w}_k)), \mathbf{w}_{k+1} - \mathbf{w}_k \rangle d\tau$$

and

$$f_m(\mathbf{w}_{k+1}) - f_m(\mathbf{w}_k) = \int_0^1 \langle \nabla f_m(\mathbf{w}_k + \tau(\mathbf{w}_{k+1} - \mathbf{w}_k)), \mathbf{w}_{k+1} - \mathbf{w}_k \rangle d\tau.$$

Denote, to simplify the expressions, $\mathbf{w}(\tau) \stackrel{\text{def}}{=} \mathbf{w}_k + \tau(\mathbf{w}_{k+1} - \mathbf{w}_k)$, then we get

$$f(\mathbf{w}_{k+1}) - f(\mathbf{w}_k) = f_m(\mathbf{w}_{k+1}) - f_m(\mathbf{w}_k) + \int_0^1 \langle \nabla f(\mathbf{w}(\tau)) - \nabla f_m(\mathbf{w}(\tau)), \mathbf{w}_{k+1} - \mathbf{w}_k \rangle d\tau$$

$$= f_m(\mathbf{w}_{k+1}) - f_m(\mathbf{w}_k) + \langle \mathbf{v}_k - \nabla f_m(\mathbf{w}_k), \mathbf{w}_{k+1} - \mathbf{w}_k \rangle$$

$$+ \int_0^1 \langle \nabla f(\mathbf{w}(\tau)) - \nabla f_m(\mathbf{w}(\tau)) - \mathbf{v}_k + \nabla f_m(\mathbf{w}_k), \mathbf{w}_{k+1} - \mathbf{w}_k \rangle d\tau.$$

Define the subproblem solved by SABER at iteration $k$ as $\phi_k(\mathbf{w}) = f_m(\mathbf{w}) + \langle \mathbf{v}_k - \nabla f_m(\mathbf{w}_k), \mathbf{w} - \mathbf{w}_k \rangle + \frac{1}{2\eta} \|\mathbf{w} - \mathbf{w}_k\|^2$. Then, it holds according to Assumption 2.1 that $\phi_k(\mathbf{w}_{k+1}) \le \phi_k(\mathbf{w}_k)$ and

$$f_m(\mathbf{w}_{k+1}) - f_m(\mathbf{w}_k) + \langle \mathbf{v}_k - \nabla f_m(\mathbf{w}_k), \mathbf{w}_{k+1} - \mathbf{w}_k \rangle \le -\frac{1}{2\eta} \|\mathbf{w}_{k+1} - \mathbf{w}_k\|^2.$$

Let us split the integral into two parts:

$$\int_0^1 \langle \nabla f(\mathbf{w}(\tau)) - \nabla f_m(\mathbf{w}(\tau)) - \mathbf{v}_k + \nabla f_m(\mathbf{w}_k), \mathbf{w}_{k+1} - \mathbf{w}_k \rangle d\tau$$

$$= \int_0^1 \langle \nabla f(\mathbf{w}(\tau)) - \nabla f_m(\mathbf{w}(\tau)) - \nabla f(\mathbf{w}_k) + \nabla f_m(\mathbf{w}_k), \mathbf{w}_{k+1} - \mathbf{w}_k \rangle d\tau$$

$$+ \langle \nabla f(\mathbf{w}_k) - \mathbf{v}_k, \mathbf{w}_{k+1} - \mathbf{w}_k \rangle.$$

The first part can be upper bounded using the data heterogeneity assumption:

$$\int_0^1 \langle \nabla f(\mathbf{w}(\tau)) - \nabla f_m(\mathbf{w}(\tau)) - \nabla f(\mathbf{w}_k) + \nabla f_m(\mathbf{w}_k), \mathbf{w}_{k+1} - \mathbf{w}_k \rangle d\tau$$

$$\le \int_0^1 \|\nabla f(\mathbf{w}(\tau)) - \nabla f_m(\mathbf{w}(\tau)) - \nabla f(\mathbf{w}_k) + \nabla f_m(\mathbf{w}_k)\| \|\mathbf{w}_{k+1} - \mathbf{w}_k\| d\tau$$

$$\le \int_0^1 \delta \|\mathbf{w}(\tau) - \mathbf{w}_k\| \|\mathbf{w}_{k+1} - \mathbf{w}_k\| d\tau = \delta \int_0^1 \tau \|\mathbf{w}_{k+1} - \mathbf{w}_k\|^2 d\tau = \frac{\delta}{2} \|\mathbf{w}_{k+1} - \mathbf{w}_k\|^2$$

$$\stackrel{\eta \le \frac{1}{4\delta}}{\le} \frac{1}{8\eta} \|\mathbf{w}_{k+1} - \mathbf{w}_k\|^2.$$

For the second part, we use Young's inequality

$$\langle \nabla f(\mathbf{w}_k) - \mathbf{v}_k, \mathbf{w}_{k+1} - \mathbf{w}_k \rangle \le 2\eta \|\nabla f(\mathbf{w}_k) - \mathbf{v}_k\|^2 + \frac{1}{8\eta} \|\mathbf{w}_{k+1} - \mathbf{w}_k\|^2.$$

Thus, combining the bounds on the integral with the initial inequalities, we obtain

$$f(\mathbf{w}_{k+1}) - f(\mathbf{w}_k) \le -\frac{1}{2\eta} \|\mathbf{w}_{k+1} - \mathbf{w}_k\|^2 + \frac{1}{8\eta} \|\mathbf{w}_{k+1} - \mathbf{w}_k\|^2 + 2\eta \|\mathbf{v}_k - \nabla f(\mathbf{w}_k)\|^2$$

$$+ \frac{1}{8\eta} \|\mathbf{w}_{k+1} - \mathbf{w}_k\|^2$$

$$= -\frac{1}{4\eta} \|\mathbf{w}_{k+1} - \mathbf{w}_k\|^2 + 2\eta \|\mathbf{v}_k - \nabla f(\mathbf{w}_k)\|^2.$$

$\square$

## A.2  Proof of Lemma 2

*Proof.* Denote by $j$ the index sampled to update $\mathbf{v}_{k+1}$, i.e., $j = m_{k+1}$. With probability $p$, we have $\mathbf{v}_{k+1} = \nabla f(\mathbf{w}_{k+1})$ if we synchronize using all clients or $\mathbf{v}_{k+1} = \frac{1}{|\widetilde{\mathcal{S}}_k|} \sum_{m \in \widetilde{\mathcal{S}}_k} \nabla f_m(\mathbf{w}_k)$ if we sample clients. In the first case, we have

$$
\begin{aligned}
&\mathbb{E}\left[\|\nabla f(\mathbf{w}_{k+1}) - \mathbf{v}_{k+1}\|^2\right] \\
&= p \cdot 0 + (1-p)\mathbb{E}\left[\|\nabla f(\mathbf{w}_{k+1}) - \mathbf{v}_k - \nabla f_j(\mathbf{w}_{k+1}) + \nabla f_j(\mathbf{w}_k)\|^2\right] \\
&= (1-p)\mathbb{E}\left[\|\nabla f(\mathbf{w}_k) - \mathbf{v}_k + \nabla f(\mathbf{w}_{k+1}) - \nabla f(\mathbf{w}_k) - \nabla f_j(\mathbf{w}_{k+1}) - \nabla f_j(\mathbf{w}_k)\|^2\right] \\
&= (1-p)\mathbb{E}\left[\|\nabla f(\mathbf{w}_k) - \mathbf{v}_k\|^2\right] \\
&\quad + (1-p)\mathbb{E}\left[2\langle \nabla f(\mathbf{w}_k) - \mathbf{v}_k, \nabla f(\mathbf{w}_{k+1}) - \nabla f(\mathbf{w}_k) - \nabla f_j(\mathbf{w}_{k+1}) - \nabla f_j(\mathbf{w}_k)\rangle\right] \\
&\quad + (1-p)\mathbb{E}\left[\|\nabla f(\mathbf{w}_{k+1}) - \nabla f(\mathbf{w}_k) - \nabla f_j(\mathbf{w}_{k+1}) - \nabla f_j(\mathbf{w}_k)\|^2\right].
\end{aligned}
$$

Since $j$ is sampled after we have produced $\mathbf{w}_{k+1}$, it is independent of $\mathbf{w}_{k+1}$, and it holds

$$
\mathbb{E}\left[\langle \nabla f(\mathbf{w}_k) - \mathbf{v}_k, \nabla f(\mathbf{w}_{k+1}) - \nabla f(\mathbf{w}_k) - \nabla f_j(\mathbf{w}_{k+1}) - \nabla f_j(\mathbf{w}_k)\rangle\right] = 0.
$$

Moreover, by second-order data heterogeneity, we have

$$
\mathbb{E}\left[\|\nabla f(\mathbf{w}_{k+1}) - \nabla f(\mathbf{w}_k) - \nabla f_j(\mathbf{w}_{k+1}) - \nabla f_j(\mathbf{w}_k)\|^2\right] \le \delta^2 \mathbb{E}\left[\|\mathbf{w}_{k+1} - \mathbf{w}_k\|^2\right].
$$

Putting the pieces together yields the claim in the first case.

In the second case, it holds

$$
\begin{aligned}
&\mathbb{E}\left[\|\nabla f(\mathbf{w}_{k+1}) - \mathbf{v}_{k+1}\|^2\right] \\
&= p\mathbb{E}\left[\left\|\nabla f(\mathbf{w}_{k+1}) - \frac{1}{|\widetilde{\mathcal{S}}_k|}\sum_{m \in \widetilde{\mathcal{S}}_k} \nabla f_m(\mathbf{w}_k)\right\|^2\right] + (1-p)\mathbb{E}\left[\|\nabla f(\mathbf{w}_{k+1}) - \mathbf{v}_k - \nabla f_j(\mathbf{w}_{k+1}) + \nabla f_j(\mathbf{w}_k)\|^2\right] \\
&\le p\mathbb{E}\left[\left\|\nabla f(\mathbf{w}_{k+1}) - \frac{1}{|\widetilde{\mathcal{S}}_k|}\sum_{m \in \widetilde{\mathcal{S}}_k} \nabla f_m(\mathbf{w}_k)\right\|^2\right] + (1-p)\|\mathbf{v}_k - \nabla f(\mathbf{w}_k)\|^2 + \delta^2\mathbb{E}\left[\|\mathbf{w}_{k+1} - \mathbf{w}_k\|^2\right] \\
&= p\frac{1}{|\mathcal{S}_k|}\mathbb{E}\left[\|\nabla f(\mathbf{w}_{k+1}) - \nabla f_m(\mathbf{w}_k)\|^2\right] + (1-p)\|\mathbf{v}_k - \nabla f(\mathbf{w}_k)\|^2 + \delta^2\mathbb{E}\left[\|\mathbf{w}_{k+1} - \mathbf{w}_k\|^2\right] \\
&\overset{(7)}{\le} p\frac{\sigma^2}{|\mathcal{S}_k|} + (1-p)\|\mathbf{v}_k - \nabla f(\mathbf{w}_k)\|^2 + \delta^2\mathbb{E}\left[\|\mathbf{w}_{k+1} - \mathbf{w}_k\|^2\right].
\end{aligned}
$$

$\square$

## A.3  Proof of Lemma 3

*Proof.* By definition $\mathbf{w}_{k+1}$ is an almost-stationary point of $\phi_k(\cdot)$ and by Assumption 2.2 we have $\mathbb{E}[\|\nabla \phi_k(\mathbf{w}_{k+1})\|^2] \le \varepsilon$. Writing the definition of $\phi_k$, we get

$$
\nabla \phi_k(\mathbf{w}_{k+1}) = \nabla f_m(\mathbf{w}_{k+1}) + \mathbf{v}_k - \nabla f_m(\mathbf{w}_k) + \frac{1}{\eta}(\mathbf{w}_{k+1} - \mathbf{w}_k),
$$

where $m = m_k$. From this equation, we obtain

$$
\begin{aligned}
&\|\mathbf{w}_{k+1} - \mathbf{w}_k\|^2 \\
&= \eta^2\|\mathbf{v}_k + \nabla f_m(\mathbf{w}_{k+1}) - \nabla f_m(\mathbf{w}_k) - \nabla \phi_k(\mathbf{w}_{k+1})\|^2 \\
&= \eta^2\|\nabla f(\mathbf{w}_{k+1}) + [\mathbf{v}_k - \nabla f(\mathbf{w}_k)] + [\nabla f(\mathbf{w}_k) - \nabla f(\mathbf{w}_{k+1}) + \nabla f_m(\mathbf{w}_{k+1}) - \nabla f_m(\mathbf{w}_k)] - \nabla \phi_k(\mathbf{w}_{k+1})\|^2.
\end{aligned}
$$

By Cauchy-Schwarz inequality, it holds $\|a_1 + a_2 + a_3 + a_4\|^2 \leq 4(\|a_1\|^2 + \|a_2\|^2 + \|a_3\|^2 + \|a_4\|^2)$ for any vectors $a_1, a_2, a_3, a_4 \in \mathbb{R}^d$. Rearranging, it also implies $\|a_4\|^2 \geq \frac{1}{4}\|a_1 + a_2 + a_3 + a_4\|^2 - \|a_1\|^2 - \|a_2\|^2 - \|a_3\|^2$, which in our case gives

$$
\begin{aligned}
\mathbb{E}\left[\|\mathbf{w}_{k+1} - \mathbf{w}_k\|^2\right] &\geq \frac{\eta^2}{4}\mathbb{E}\left[\|\nabla f(\mathbf{w}_{k+1})\|^2\right] - \eta^2 \mathbb{E}\left[\|\mathbf{v}_k - \nabla f(\mathbf{w}_k)\|^2\right] \\
&\quad - \eta^2 \mathbb{E}\left[\|\nabla f(\mathbf{w}_k) - \nabla f(\mathbf{w}_{k+1}) + \nabla f_m(\mathbf{w}_{k+1}) - \nabla f_m(\mathbf{w}_k)\|^2\right] - \eta^2\|\nabla\phi_k(\mathbf{w}_{k+1})\|^2 \\
&\overset{(2)}{\geq} \frac{\eta^2}{4}\mathbb{E}\left[\|\nabla f(\mathbf{w}_{k+1})\|^2 - \|\mathbf{v}_k - \nabla f(\mathbf{w}_k)\|^2 - \delta^2\|\mathbf{w}_{k+1} - \mathbf{w}_k\|^2 - \varepsilon\right].
\end{aligned}
$$

Notice that $\|\mathbf{w}_{k+1} - \mathbf{w}_k\|^2$ appears in both sides, so we can rearrange and divide by $1 + \eta^2\delta^2$:

$$
\begin{aligned}
\mathbb{E}\left[\|\mathbf{w}_{k+1} - \mathbf{w}_k\|^2\right] &\geq \frac{1}{1 + \eta^2\delta^2}\mathbb{E}\left[\frac{\eta^2}{4}\|\nabla f(\mathbf{w}_{k+1})\|^2 - \eta^2\|\mathbf{v}_k - \nabla f(\mathbf{w}_k)\|^2 - \eta^2\varepsilon\right] \\
&\overset{\eta \leq \frac{1}{4\delta}}{\geq} \frac{16}{17}\mathbb{E}\left[\frac{\eta^2}{4}\|\nabla f(\mathbf{w}_{k+1})\|^2 - \eta^2\|\mathbf{v}_k - \nabla f(\mathbf{w}_k)\|^2 - \eta^2\varepsilon\right] \\
&\geq \mathbb{E}\left[\frac{\eta^2}{5}\|\nabla f(\mathbf{w}_{k+1})\|^2 - \eta^2\|\mathbf{v}_k - \nabla f(\mathbf{w}_k)\|^2 - \eta^2\varepsilon\right].
\end{aligned}
$$

$\square$

### A.4 Proof of Theorem 1

*Proof.* Recall that we define a Lyapunov function

$$
\mathcal{L}_k \overset{\text{def}}{=} f(\mathbf{w}_k) + c\|\mathbf{v}_k - \nabla f(\mathbf{w}_k)\|^2,
$$

where we will choose $c > 0$ later in the proof. Lemmas 1 and 2 already bound the first and the second terms in $\mathcal{L}_{k+1}$ correspondingly, giving us the following recursion:

$$
\begin{aligned}
\mathbb{E}\left[\mathcal{L}_{k+1}\right] &\leq f(\mathbf{w}_k) + \mathbb{E}\left[-\frac{1}{2\eta}\|\mathbf{w}_{k+1} - \mathbf{w}_k\|^2 + 2\eta\|\nabla f(\mathbf{w}_k) - \mathbf{v}_k\|^2\right] \\
&\quad + c(1-p)\|\nabla f(\mathbf{w}_k) - \mathbf{v}_k\|^2 + c\delta^2\mathbb{E}\left[\|\mathbf{w}_{k+1} - \mathbf{w}_k\|^2\right] \\
&= \mathcal{L}_k - c\|\nabla f(\mathbf{w}_k) - \mathbf{v}_k\|^2 + \left(c\delta^2 - \frac{1}{2\eta}\right)\mathbb{E}\left[\|\mathbf{w}_{k+1} - \mathbf{w}_k\|^2\right] \\
&\quad + (2\eta + c(1-p))\|\nabla f(\mathbf{w}_k) - \mathbf{v}_k\|^2 \\
&= \mathcal{L}_k + \left(c\delta^2 - \frac{1}{2\eta}\right)\mathbb{E}\left[\|\mathbf{w}_{k+1} - \mathbf{w}_k\|^2\right] + (2\eta - cp)\|\nabla f(\mathbf{w}_k) - \mathbf{v}_k\|^2.
\end{aligned}
$$

Let us set $c = \frac{3\eta}{p}$ to make the last term negative. Then, we obtain

$$
\begin{aligned}
\mathbb{E}\left[\mathcal{L}_{k+1}\right] &\leq \mathcal{L}_k + \left(\frac{3\eta\delta^2}{p} - \frac{1}{2\eta}\right)\mathbb{E}\left[\|\mathbf{w}_{k+1} - \mathbf{w}_k\|^2\right] - \eta\|\nabla f(\mathbf{w}_k) - \mathbf{v}_k\|^2 \\
&\overset{\eta \leq \frac{\sqrt{p}}{4\delta}}{\leq} \mathcal{L}_k - \frac{1}{4\eta}\mathbb{E}\left[\|\mathbf{w}_{k+1} - \mathbf{w}_k\|^2\right] - \eta\|\mathbf{v}_k - \nabla f(\mathbf{w}_k)\|^2 \\
&\overset{(4)}{\leq} \mathcal{L}_k - \frac{\eta}{20}\mathbb{E}\left[\|\nabla f(\mathbf{w}_{k+1})\|^2\right] + \frac{\eta}{4}\|\mathbf{v}_k - \nabla f(\mathbf{w}_k)\|^2 - \eta\|\mathbf{v}_k - \nabla f(\mathbf{w}_k)\|^2 + \frac{\eta}{4}\varepsilon \\
&\leq \mathcal{L}_k - \frac{\eta}{20}\mathbb{E}\left[\|\nabla f(\mathbf{w}_{k+1})\|^2\right] + \frac{\eta}{4}\varepsilon.
\end{aligned}
$$

Recurring this to $\mathcal{L}_0 = f(\mathbf{w}_0) + c\|\mathbf{v}_0 - \nabla f(\mathbf{w}_0)\|^2 = f(\mathbf{w}_0)$, we get

$$
\frac{1}{K}\sum_{k=1}^K \mathbb{E}\left[\|\nabla f(\mathbf{w}_k)\|^2\right] \leq \frac{20}{\eta K}\left(\mathcal{L}_0 - \mathbb{E}\left[\mathcal{L}_K\right] + K\frac{\eta}{4}\varepsilon\right) \leq \frac{20(f(\mathbf{w}_0) - f_*)}{\eta K} + 5\varepsilon,
$$

where we used the fact that $\mathcal{L}_K = f(\mathbf{w}_K) + c\|\mathbf{v}_K - \nabla f(\mathbf{w}_K)\|^2 \geq f(\mathbf{w}_k) \geq f_*$. $\square$

### A.5 Proof of Theorem 3

*Proof.* We proceed with the same steps as in the proof of Theorem 1 except that we take into account the extra term that appeared in the proof of Lemma 2:

$$\mathbb{E}\left[\|\nabla f(\mathbf{w}_{k+1}) - \mathbf{v}_{k+1}\|^2\right] \leq p\frac{\sigma^2}{|\mathcal{S}_k|} + (1-p)\|\mathbf{v}_k - \nabla f(\mathbf{w}_k)\|^2 + \delta^2\mathbb{E}\left[\|\mathbf{w}_{k+1} - \mathbf{w}_k\|^2\right].$$

Using the same Lyapunov function as before, we obtain by plugging-in the result above

$$\mathbb{E}\left[\mathcal{L}_{k+1}\right] \leq \mathcal{L}_k + \left(c\delta^2 - \frac{1}{2\eta}\right)\mathbb{E}\left[\|\mathbf{w}_{k+1} - \mathbf{w}_k\|^2\right] + (2\eta - cp)\|\nabla f(\mathbf{w}_k) - \mathbf{v}_k\|^2 + cp\frac{\sigma^2}{|\mathcal{S}_k|}.$$

Setting $c = \frac{3\eta}{p}$ and substituting the other bounds, we derive

$$\mathbb{E}\left[\mathcal{L}_{k+1}\right] \leq \mathcal{L}_k - \frac{\eta}{20}\mathbb{E}\left[\|\nabla f(\mathbf{w}_{k+1})\|^2\right] + \frac{\eta}{4}\varepsilon + 3\eta\frac{\sigma^2}{|\mathcal{S}_k|}.$$

Using the theorem assumption that $|\mathcal{S}_k| = C$ for all $k$ and rearranging, we further obtain

$$\frac{1}{K}\sum_{k=1}^{K}\mathbb{E}\left[\|\nabla f(\mathbf{w}_k)\|^2\right] \leq \frac{20(f(\mathbf{w}_0) - f_*)}{\eta K} + 5\varepsilon + 60\frac{\sigma^2}{|\mathcal{S}_k|}.$$

$\square$

