# OpenReview forum: "Federated Learning Under Second-Order Data Heterogeneity"
_TMLR — Rejected by TMLR_

### Review · Reviewer_28HN · 2024-05-23

**Summary Of Contributions:**

This paper introduces SABER, an algorithm designed to address second-order data heterogeneity in Federated Learning (FL). SABER mitigates client drift by incorporating global update direction estimates and regularization into local objectives, maintaining a stateless nature, and supporting partial client participation. Theoretically, SABER achieves communication complexity of $ O(\delta \epsilon^{-2} \sqrt{M}) $ for non-convex problems and demonstrates linear convergence under the $\mu$-Polyak-Łojasiewicz condition with complexity $ O((\frac{\delta}{\mu \sqrt{M}} + M) \log \frac{1}{\epsilon}) $. Empirical evaluations on CIFAR-10 and FEMNIST datasets show that SABER outperforms FedAvg, FedProx, and SCAFFOLD in heterogeneous data settings, achieving faster convergence and higher accuracy.

**Audience:**

Yes

**Claims And Evidence:**

Yes

**Requested Changes:**

Please try to discuss the Weaknesses and may make some changes with following aspects:
- Expand on the motivation behind the design of SABER, explaining why the combination of FedProx and SCAFFOLD elements is particularly effective for second-order data heterogeneity.
- Provide a more detailed comparison with recent related works, such as Lin et al. (2023). Discuss the similarities and differences in greater depth to contextualize SABER's contributions.

Dachao Lin, Yuze Han, Haishan Ye, and Zhihua Zhang. Stochastic Distributed Optimization under Average Second-order Similarity: Algorithms and Analysis. arXiv preprint arXiv:2304.07504, 2023.

**Strengths And Weaknesses:**

Strengths:
-The issue addressed in this paper is well-founded.
- The method appears to be sound, and I appreciate the theoretical bounds provided for both general cases and in the case of $\mu$-PL.
- The theory is supported by extensive numerical experiments.

Weaknesses:
- The local objective function constructed by the authors is an intuitive combination of FedProx and SCAFFOLD, with similar ideas found in Lin et al. (2023).
- The SABER algorithm design is not particularly surprising, as it uses a strategy akin to SVRG to address data heterogeneity.
- The main proof techniques are standard in federated learning, with no concrete improvements over existing state-of-the-art results.

Dachao Lin, Yuze Han, Haishan Ye, and Zhihua Zhang. Stochastic Distributed Optimization under Average Second-order Similarity: Algorithms and Analysis. arXiv preprint arXiv:2304.07504, 2023.

---

### Review · Reviewer_yoPJ · 2024-07-23

**Summary Of Contributions:**

This article proposes a federated learning method in the non-convex optimization setting when the client data is heterogeneous. Under a second-order data heterogeneity assumption, the convergence rate of the method is given. This rate is further improved to be linear under an extra PL condition. The performance of the method is also evaluated numerically on logistic regression problems for image classification.

**Audience:**

Yes

**Claims And Evidence:**

No

**Requested Changes:**

-	Clarify the points in the above section.
-	Complete your main theorem with the full assumptions made, e.g. in Theorem 3, you might want to add Assumption 4 into the statement.
-	Explain what is the t on line 5 of algorithm 2. Should it be k instead?

**Strengths And Weaknesses:**

Strength: the proposed method SABER in section 3 seems to be original. By solving a convex sub-problem on line 7 of Algorithm 1, the convergence theory for non-convex objective functions are obtained.

Weakness: writing clarity and soundness of both theoretical and numerical results.
- It is unclear how do you define the FedAvg method in terms of the \approx , I think the \approx in the equation on page 5 about FedAvg is not clear. Similarly on page 5 the equation about E(v_k), is there any typo about grad f (w_k) which should have been grad_m f (w_k)?
- In terms of theory, the definition of the complexity of the proposed method does not seem to take into account of the computational time of the sub-problem in Section 3.1. Also, it is not clear why under the condition about the learning rate eta in Theorem 1, the sub-problem becomes convex. As this is a key idea of the method, I think it is important to clarify these points.
- In terms of experiments, are you considering the logistic regression problem which is a convex by its nature? It is unclear from the text whether you are considering non-convex problems which should have been the focus of the article.

---

### Review · Reviewer_Z73Q · 2024-07-25

**Summary Of Contributions:**

The authors consider the heterogenous federated learning task of minimizing an objective $f(w) = \frac{1}{M} f_m(w)$, where $f_m$ are client-specific objectives. The work assume that the second order heterogeneity is bounded among the clients, ie the difference between the client's Hessians and the Hessian of the joint objective $f$ is bounded in Spectral norm by $\delta$.

The authors focus on the non-convex setting, with and without a PL condition. They study a model where there is infequent full syncronization (a $p$ fraction of rounds) and every round a randomly chosen single client mnimizes a certain objective. They develop a novel algorithm SABER which can be viewed as a combination of SCAFFOLD and FedProx. One advantage of their approach, over more standard FL approaches is that it is *stateless*, that is, if a client $m$ is not involved in a certain round, it does not need to store anything.

Without a PL condition, they show that for $M$ clients, after $\sqrt{M}\delta \epsilon^{-2}$ rounds, with $p = 1/M$, SABER converges to $\||\nabla f(x)\||^2 \leq \epsilon$. Assuming the objective is $\mu$-PL, SABER with $p = 1/M$ converges in $(\delta \sqrt{M}/\mu + M)\log(1/\epsilon)$ rounds.

**Audience:**

Yes

**Broader Impact Concerns:**

I have no concerns

**Claims And Evidence:**

Yes

**Requested Changes:**

**Critical**
- Expositon:
  - The discussion in the abstract and introduction should clarify and isolate the most improtant axis on which improvement is made in this work. (Weaker assumptions? Communication complexity? Practicality of algorithm?)
  - A quantitative table comparing this result to others in the non-convex heterogeneous setting.
- Technical Results:
  - An extention of their theorems to give a full gradient-oracle or stochastic gradient oracle complexity, that accounts for the complexity of minimizing the local objective under standard assumptions. (The authors have a short discussion of this on the bottom of page 6, but it should be made formal and quantitative. This can then be compared to other results in the literature).
  - (Not 100% critical, but highly recommended) A version of SABER should be analyzed which is more similar to the traditional intermittend communicaiton setting of FedAvg for example, were all clients perfom the local optimization at each round. In essence, Alg 1 of SABER is similar to a partial participlation setting where exactly 1 client is chosen in each round. So understanding the full participation setting would be natural and useful for comparison to the literature.


**Additional**
- Results in the *convex* setting for comparison purposes (not just strongly convex or PL). I don't think this is critical since the main focus of the paper is on non-convex and non-PL, but having PL results but no results for the convex settings (or as $\mu \rightarrow 0$) seems strange.

**Strengths And Weaknesses:**

**Strengths**
- The technical innovation of SABER is is clear, intiutive, well-explained and properly compared to related algorithms. The explanations on page 5 and first half of page 6 are very clear and helpful.
- Working with the Hessian similarity condition is of interest, and likely of more practical value than other heterogeneity assumptions in the literature.
- The algorithm is stateless, which may be more practical in real implementation settings.
- The algorithm is shown to perform well empirically on some image datasets.

**Weaknesses**
- The main weakness of this paper is that the quantitative comparisons to the existing literature are lacking, and it is not clear if (or how) the result here is state of the art. The PL setting is non-standard, so I will focus on the non-convex non-PL setting.
  - The work "Towards Optimal Communication Complexity in Distributed Non-Convex Optimization" by Patel et al. is not mentioned by the authors, and provides very important results in this realm. Perhaps most relevant is their Theorem 3.3, which achieves a communication complexity propotional to $\delta^{⅔}\epsilon^{-3/2}$ (most significant term for small $\epsilon$, see their Table 2) . At least in dependence on $\epsilon$, this seems to improve upon the authors work on SABER, which achieves a communicaiton complexity proportional to $\epsilon^{-2}$. The work "Mime: Mimicking Centralized Stochastic Algorithms in Federated Learning." is also a relevant comparison point which achieves similar $\epsilon^{-3/2}$ dependence.
  - The authors in passing mention several other similar works (for instance FedPAGE, Scaffold), but never give concrete quantitative bounds to compare. Most importantly works with similiar heterogeneity assumptions should be compared quantitatively.
  - Its not clear what is the single most imporant contribution of this work.  (Weaker assumptions? Communication complexity? Practicality of algorithm?). For example on page 5 (bottom) the authors discuss how they do not assume smoothness, even though this is a very standard assumption. Is this a main contribution? It was not dicussed earlier in the introduction or abstract.
  -

---

### Review · Reviewer_TbnV · 2024-07-26

**Summary Of Contributions:**

TLDR: Well written paper which closes important gaps in FL convergence analysis, but overstates its contributions



The authors propose a novel federated learning method (SABER) and analysis that shows a $O(\delta \epsilon^{-2} \sqrt{M})$ communication complexity for non-convex functions. They also provide improved linear convergence rates when the PL condition is satisfied. This improves the previous known rate for non-convex FL by [FedPage](https://arxiv.org/abs/2108.04755) who showed $O(L \epsilon^{-2} \sqrt{M})$ i.e. they replace the dependence on smoothness term $L$ with second-order heterogeneity $\delta$ which may be significantly smaller. Their results can also be viewed as extending the analysis of [SVRS](https://arxiv.org/abs/2304.07504) who show similar improvements with second-order heterogeneity for convex functions.

Finally, the algorithm itself is very reminiscent of [FedDyn](https://openreview.net/forum?id=B7v4QMR6Z9w) [Acar et al. 2023] i.e. it uses a bias correction as in SCAFFOLD and an additional explicit regularization term. The key difference is the randomized number of local steps. This is similar to the difference between Scaffnew and SCAFFOLD---the randomized number of local steps facilitates cleaner and sharper convergence analysis.

**Audience:**

Yes

**Broader Impact Concerns:**

none.

**Claims And Evidence:**

Yes

**Requested Changes:**

See weaknesses above.

**Strengths And Weaknesses:**

**Strengths**

1. *Second-order heterogeneity:* SABER addresses second-order data heterogeneity, a less-explored aspect in federated learning. It closes an important gap in the literature -- optimal non-convex communication complexity. While the paper itself does not discuss lower-bounds, the lower bound of [[Fang et al. 2018](https://proceedings.neurips.cc/paper_files/paper/2018/hash/1543843a4723ed2ab08e18053ae6dc5b-Abstract.html)] together with techniques in [[Arjevani et al. 2015](https://arxiv.org/abs/1506.01900)] implies nearly matching conditional communication complexity lower-bounds.

2. *Theoretical Guarantees*: The paper offers a well-written rigorous theoretical analysis, demonstrating improved communication complexity for non-convex problems and faster linear convergence under Polyak-Łojasiewicz conditions.

3. *Empirical Validation:* Extensive experiments on logistic regression and image classification tasks show that SABER outperforms existing baselines, particularly in highly heterogeneous data settings.

4. *Partial Participation:* The algorithm supports partial participation, making it more practical for real-world federated learning deployments.


**Weaknesses**

1. *Better comparison with existing results:* FedDyn is very closely related to the current method and should be discussed. Further, the relation to FedPage should be discussed much more prominently, ideally starting from the introduction itself. It seems like the authors were made aware of FedPage late into their writing. While understandable, I highly recommend the authors rewrite their introduction to reflect FedPage and SVRS (perhaps along the narrative in the summary above).

2. *Lowerbounds:* The paper can also be made more complete by discussing lower bounds and the optimality of their rates as stated above.

3. *Sateleness:* FedDyn is also a stateless algorithm. SCAFFOLD can also be made stateless (see Appendix A of [[Yu et al. 2022](https://arxiv.org/abs/2207.06343)]). So I recommend removing the uniqueness of statelessness as a contribution, or at least de-emphasizing it.

---

### Decision · Action_Editor_9x5o · 2024-08-23

**Recommendation:** Reject

**Comment:**

This paper is a somewhat borderline case, but I believe that all reviewers (including the ones who leaned accept) had substantial, but concrete, suggestions for how to improve the work. These suggestions all seem useful, constructive, and I believe would put the paper in a spot to be clearly accepted.

I wish to emphasize that this work is promising - the reviewers all seemed interested in the direction, topic, and cared about the actual analysis underlying it. I would strongly urge the authors to resubmit, but I think that the case for the paper's contributions is not quite convincing enough to warrant acceptance as is.

**Audience:**

I wish to emphasize that the reviewers were pretty agreed that the types of contributions in this work are clearly of interest to audiences. This is why I would like to emphasize to the authors that I believe this work should be resubmitted - but after thinking about and working through some of the feedback given by the reviewers.

**Claims And Evidence:**

This part of the work is where nearly all reviewers had critiques of the work. There were multiple areas where the reviewers suggested that the paper needs to revise its claims, including:

* **Statelessness:** As discussed by reviewer TbnV, there are already federated learning methods that can deal with second-order data heterogeneity, and are stateless (e.g. FedDyn) or can be made stateless. Note that while FedDyn has not (to the best of my knowledge) been shown to converge under second-order heterogeneity, the similarities between SABER and FedDyn suggest that such an analysis could go through.
* **State-of-the-art communication complexity:** As Reviewer Z73Q, this may not be the case. Mime may match this, and work by Patel et al., who may obtain improved communication-complexity over SABER. Additionally, and as is backed up by Reviewers TbnV and 28HN, the work doesn't give quantitative comparisons between very relevant methods such as FedPAGE and SCAFFOLD.
* **Convergence under Polyak-Lojasiewicz conditions:** There was some confusion among reviewers about this contribution. Reviewer Z73Q mentioned that a convergence in convex settings might be useful for comparison to other works. More generally, I found the distinction between the convergence of SABER in PL conditions a bit hard to parse. The paper states that [unlike SVRP and SVRS, our method does not require the objective to be convex]. Is this a fundamental limitation in methods such as SVRP and SVRS, or could the analysis in this work be adapted to that setting?

Reviewers TbnV explicitly mentioned comparisons to lower bounds as a suggestion for a way to more clearly support the contributions of this work, highlighting a way to show that SABER is nearly optimal. I also believe that such a comparison would go a long way to addressing the feedback of Reviewers 28HN and Z73Q.

There is some evidence for the various claims given in the work, but upon re-reading the paper after reading the reviewer feedback, I think that this work may need some changes in order to accurately showcase how the convergence of SABER improves over other related methods, and to refine the message of what the core contributions are.

**Resubmission Of Major Revision:**

The authors may consider submitting a major revision at a later time.